# Simulation of Mushroom Nanostructures with Ag Nanoparticles for Broadband and Wide-Angle Superabsorption

**Jinshuang Wu** [1], **Mingzhao Ouyang** [1,2,*], **Bowei Yang** [1] **and Yuegang Fu** [1,2,*]

1    School of Optoelectronic Engineering, Changchun University of Science and Technology, Changchun 130022, China

2    Key Laboratory of Optoelectronic Measurement and Optical Information Transmission Technology of Ministry of Education, Changchun University of Science and Technology, Changchun 130022, China

*    Correspondence: oymz@cust.edu.cn (M.O.); fuyg@cust.edu.cn (Y.F.); Tel.: +86-0431-8558-3369 (Y.F.)

**Abstract:** Metal nanoparticles (NPs) concentrate the energy of incident photons through plasmon resonance excitation, which allows scattering into a substrate with a high refractive index, and the radiated energy from this excitation significantly increases the optical absorption of the substrate. In this work, the effect of Ag NPs on the absorption capacity of mushroom-nanostructured Si metasurfaces was analyzed using the finite-difference time-domain method. It was observed that the absorbance in the metasurfaces with Ag NPs increased from 90.8% to 98.7% compared with nanostructured Si metasurface without NPs. It was shown that the plasmon resonance effect of Ag NPs enlarged the range of the FP cavity by about 10 times, and the electric field strength $|E|^2$ increased by about four times through the combination of Ag NP and Si absorbers. Meanwhile, the effect of randomly distributed nanostructures on the absorption properties of Si metasurfaces was simulated. Additionally, the nanostructured surface with Ag NPs was insensitive to angle, which encourages the design of broadband and wide-angle superabsorption nanostructures.

**Keywords:** plasmon resonance; finite difference time domain method; superabsorption; subwavelength structure

## 1. Introduction

The demand for light energy utilization grows along with the popularity of light absorbers in various research fields [1–3], especially in photovoltaic devices [4–6], photodetection [7,8], thermal imaging [9,10], and solar cells [11,12]. In order to achieve broadband light absorption from the visible to the terahertz frequencies, the use of metamaterials and subwavelength nanostructures has greatly increased [13–16].

Many metamaterial absorbers have perfect absorption and are insensitive to the incident angle and polarization [17,18]. A typical metamaterial absorber is composed of a metal-dielectric multilayer film, which achieves broadband absorption through impedance matching and resonance [19–21]. Kim et al. first introduced hyperuniform disordered patterns to a metal-insulator-metal gap plasmon metasurface, demonstrated enhanced wideband light absorption in the visible and near-infrared spectral region, and revealed the origins of the two resonance modes of the gap plasmon metasurface structure [22]. Furthermore, the subwavelength structure can capture the incident light, and the combination of the subwavelength structure and metallic particles can increase the plasmon resonance, which further supports the broadband absorption [23–25]. Moreover, Zhou proposed a wide-band bidirectional visible-light absorber based on a quasi-periodic nanocone array coated with a dielectric-loaded Au monolayer, which is capable of simultaneously absorbing light from both the front and rear surfaces. They obtained an average front absorption of 87.4% and a corresponding rear absorptivity of 76.2% in the 300–700 nm range [26]. However, the above methods have issues such as limited resonance bandwidth, ther-

mal mismatch of multilayer materials, complex processing technology of subwavelength structures, and high cost.

These issues were addressed in our previous research by fabricating a silicon metasurface with mushroom structures, using an effective method based on metal thermal annealing and inductively coupled plasma (ICP) processes. The effect of the diameter of the hemispherical Ag NP and the height of the waist-shape structure on the absorption properties of the silicon metasurface was analyzed using the FDTD method. The absorptance of the prepared silicon metasurface was ~98% in the wavelength range from 360 to 2000 nm. It was proved that the structure had nearly perfect superabsorption over a broadband range from UV to NIR. However, the physical mechanism behind this superabsorption was reasonably explained in our previous work [27]. In this work, the finite difference time domain (FDTD) method was used to analyze and simulate the influence of Ag nanoparticles (NPs) on the absorption of Si metasurfaces. The broadband absorption levels of nanostructured Si surfaces with and without Ag NPs were compared. In addition, the effect of randomly distributed nanostructures on the absorption properties of Si metasurfaces was simulated via FDTD. Furthermore, the absorbance of nanostructured Si metasurfaces with Ag NPs for varying incidence angles was studied to test its wide-angle performance for various applications.

## 2. Materials and Methods

### 2.1. Model

Figure 1a shows a schematic diagram of the proposed nanostructure, which is composed of three different layers. The top is made up of Ag NPs to enhance the absorption efficiency owing to the coupling and interaction of surface plasmons. The middle part is a waist-shaped nanostructure to reduce the reflection of incident light, and the bottom layer is the Si substrate.

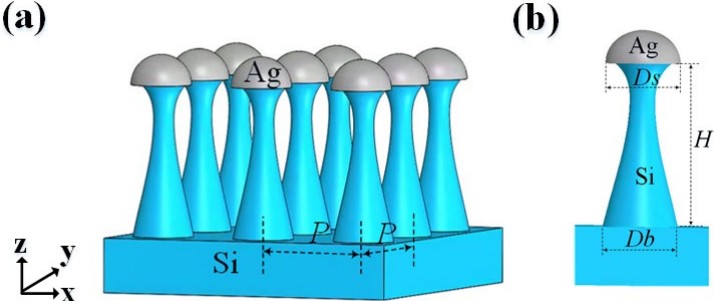

**Figure 1.** (**a**) Schematic diagram of the proposed mushroom-nanostructured arrays; (**b**) 2D view of the nanostructure with Ag NPs in a unit cell.

Moreover, Figure 1b shows the 2D unit cell view of the proposed nanostructure with Ag NPs. The main structural parameters of the unit cell are the period of the structure *P*, the Ag NP diameter *Ds*, the height *H*, and the bottom diameter *Db* of the waist-shaped nanostructure.

### 2.2. Simulation

The optical properties of the silicon metasurface with and without Ag NPs were simulated using the commercial software Lumerical FDTD Solutions. The FDTD simulation model was built in an *x-y-z* axis space rectangular coordinate system, which included the silicon substrate, waist-shaped nanostructure array, air layer, source, and monitor. Both the reflected and transmitted waves were monitored to calculate the reflectance (*R*) and transmittance (*T*). The absorbance (*A*) was calculated as $A = 1 - R - T$. Moreover, the electric field distribution on the unit cell array was also obtained from the simulations at different incident wavelengths. The FDTD Solutions software used a plane wave source with a polarization direction along the *x* axis, and the boundary conditions of the *x* and

*y* axis were periodic, where the perfectly-matched layer boundary condition was applied on the *z* axis. The Bloch boundary condition was used for oblique incidence light. In order to obtain accurate simulation results, the mesh override regions dx, dy, and dz were 0.01 µm, the simulation time was 5000 fs, and the boundary layer was 12 in FDTD. The materials Si and Ag were taken from Si (Silicon)—Palik and Ag (Siliver)—Palik (0–2 µm) in the material library, respectively. The parameters of the simulation model were set as $P = 800$ nm, $Db = Ds = 600$ nm, and $H = 1000$ nm, and the thickness of the Si substrate was 2 µm.

### 2.3. Experiment

The expected mushroom-nanostructures were prepared with metal annealing and ICP processes. The process diagram of the nanostructure preparation is shown in Figure 2. First, the Si samples were cleaned by the Radio Corporation of America (RCA) process, and a 50-nm-thick Ag film was deposited on the substrates via sputter coating (Alliance Concept, DP650, Annecy, France). Thereafter, the samples were placed in a muffle furnace for rapid thermal annealing at 350 °C for 3 min, thereby forming a mask of Ag NPs. Then the nanostructures were prepared using Ag NPs masks via an ICP process. The etching parameters were maintained at 300 W and 20 W of upper source and lower electrode power, respectively. A mixture of 30 sccm $CHF_3$ and 10 sccm $SF_6$ was used as the etching gas at a pressure of 0.5 Pa for 80 s. In addition, the samples without Ag NPs were obtained by removing residual Ag NPs with an etching solution.

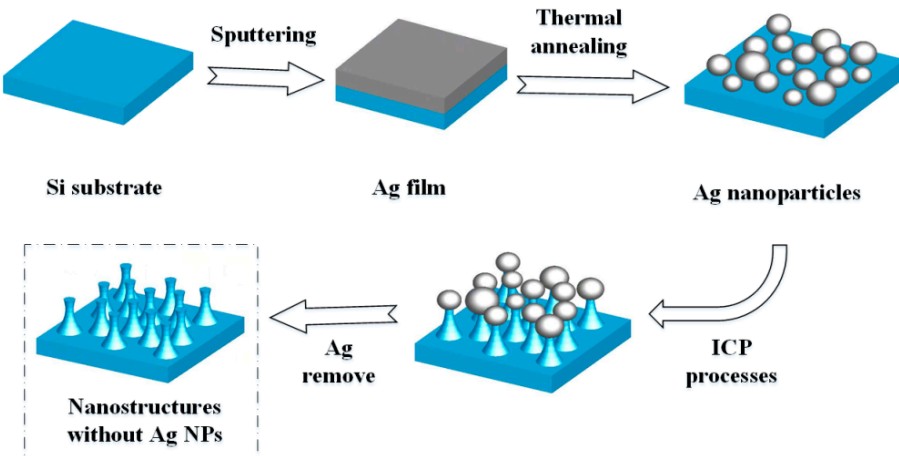

**Figure 2.** Process diagram for preparing the mushroom-nanostructures via metal thermal annealing and ICP processes.

### 2.4. Characteristics

The structured morphology was observed using scanning electron microscopy (SEM, Quanta 250, FEI, Hillsboro, OR, USA). Then, the optical reflectance and transmittance were measured using a universal scanning spectrophotometer (EssentOptics, PHOTON RT, Minsk, Belarus). The wavelength range of the spectrometer source was set from 360 to 2000 nm. The slit width was set at 230 µm.

## 3. Results and Discussion

A 3 × 3-unit cell array was established as the simulation model, as shown in Figure 1a, in order to investigate the effect of Ag NPs on the absorption of Si metasurface with nanostructures. The absorbance, reflectance, and transmittance for the simulation model without and with Ag NPs were calculated via FDTD Solutions.

### 3.1. The Effect of Ag NPs on the Absorption of Si Metasurfaces

Figure 3a shows the optical absorption properties of Ag NPs on Si substrate (before the ICP process), which had absorption peaks at 0.42 μm, 0.82 μm, 1.12 μm and 1.42 μm. This was due to the fact that the incident light coupled with the different positions of the Ag NPs to form surface plasmon resonance, which reduced the reflection of specific wavelengths and thus enhanced the absorption. As shown in Figure 3b, the nanostructured Si metasurface without Ag NPs had an average absorbance of 92.3% and an average reflectance of 7% in the wavelength range of 400–2000 nm. Meanwhile, Figure 3c shows that the nanostructured Si metasurface with Ag NPs had a higher average absorbance of 98.2%, and the average reflectivity was also just 1.6% lower than that of the nanostructures without Ag NPs. This could be attributed to the transition behavior of free electrons induced by the addition of metallic Ag NPs, and the plasmon resonance excitation of the gap plasmon mode or the surface plasmon mode, which concentrated the energy of incident photons into the structure. Therefore, based on superabsorption, the nanostructured Si metasurface with Ag NPs was confirmed to exhibit higher absorbance characteristics in the wavelength range of 400–2000 nm compared to that without Ag NPs.

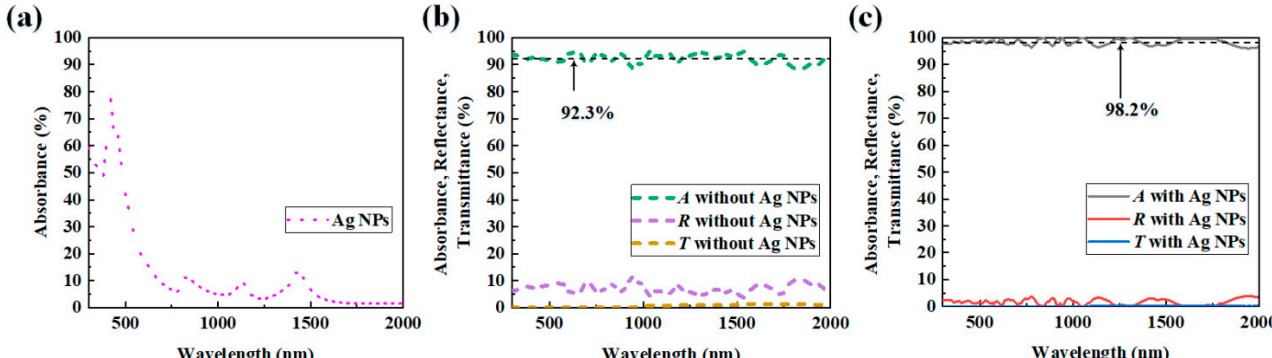

**Figure 3.** Simulated optical properties of (**a**) Ag NPs on Si substrate (before ICP process) and nanostructured Si metasurface (**b**) without and (**c**) with Ag NPs (*A*-Absorbance, *R*-Reflectance, *T*-Transmittance, the black dotted line corresponding to the average absorbance).

Nanostructures without and with Ag NPs were prepared via the processes shown in Figure 2. Furthermore, Figure 4 shows the cross-sectional SEM images of the nanostructures without and with Ag NPs after the ICP process. The nanostructure arrays had uniform dimensions, with a height of 1080 nm and a pitch of 230 nm between two adjacent nanostructures, which was achieved using thermal annealing and etching. The experimental etching structure is consistent with the proposed simulation model. The pitch and height of the nanostructures could be experimentally controlled by changing the etching time and gas flow.

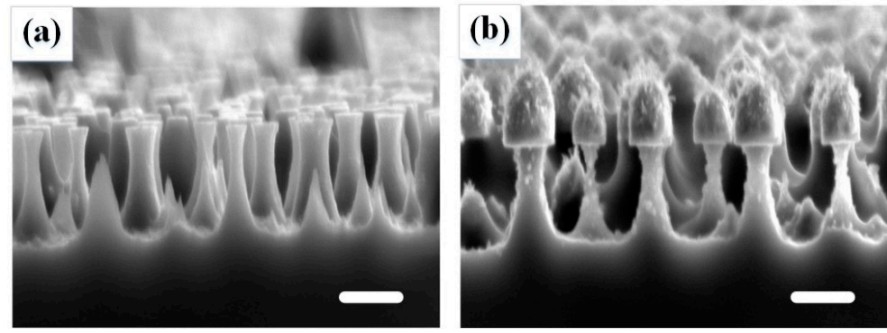

**Figure 4.** Cross-sectional SEM images of the nanostructures (**a**) without Ag NPs and (**b**) with Ag NPs after the ICP process. Scale bar corresponds to 500 nm.

Moreover, optical reflectance (*R*) and transmittance (*T*) were measured using a universal scanning spectrophotometer, and the absorbance (*A*) was calculated as $A = 1 - R - T$. Figure 5 shows the measured reflectance, transmittance, and absorbance spectra of the nanostructured Si metasurface without and with Ag NPs after the ICP etching process. At wavelengths greater than 1400 nm, the nanostructured Si metasurface without Ag NPs showed absorbance values less than 90%, while the average reflectance was higher at approximately 9%, as shown in Figure 5a. Consistent with the simulated results, the absorption of the nanostructured Si metasurface with Ag NPs also significantly improved. The average absorption was 98.7% in the wavelength range of 400–2000 nm, as shown in Figure 5b, showing a near perfect broadband absorption effect.

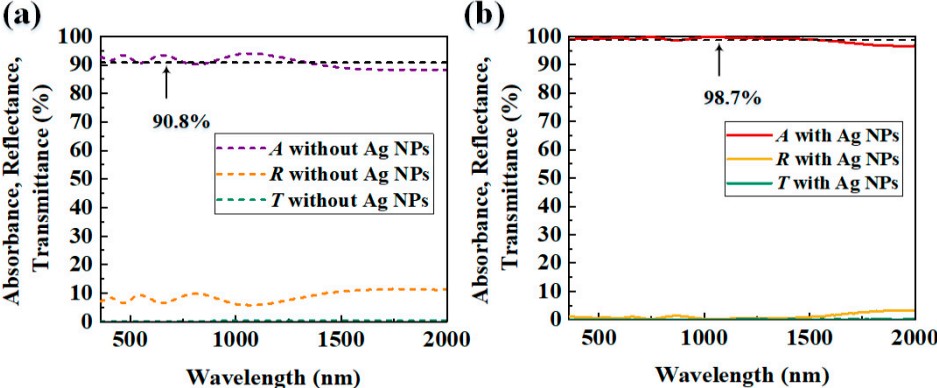

**Figure 5.** Measured reflectance, transmittance, and absorbance spectra of the nanostructured Si metasurface (**a**) without and (**b**) with Ag NPs after etching via the ICP process (the black dotted line corresponding to the average absorbance).

### 3.2. The Effect of Nanostructured Si Surface with Ag NPs on Wide-Angle Superabsorption

We also simulated the influence of Ag NPs on the absorption ability of the mushroom-structured Si metasurface. Figure 6a shows the angular dependence of the absorption-wavelength curves for incident angles ranging from 0 to 60°. Within 0 to 30°, the absorption curves of the nanostructures with Ag NPs were almost indistinguishable, and all absorbance exceeded 98.6% in the whole spectrum range of 400–2000 nm, which indicated insensitivity to the angle of incidence. However, as the incident angle increased, the absorption weakened. At 60°, the absorption capacity was significantly smaller than in the 0–45° range at wavelengths of 1200–2000 nm, and the minimum absorbance was only 96%. However, this was still higher than the average absorptivity of 92.3% for the nanostructures without Ag NPs under normal incidence, as shown in Figure 3a.

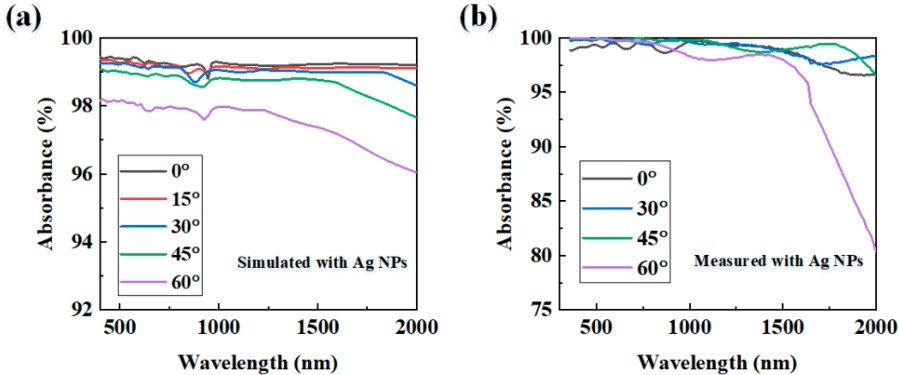

**Figure 6.** Absorbance spectrum of the nanostructured Si metasurface at different angles of incidence from 0° to 60° with an interval of 15°; (**a**) simulated and (**b**) measured with Ag NPs.

To verify the performance of the large-angle superabsorption on the fabricated nanostructured Si metasurface with Ag NPs, the angle-dependent absorption spectra were measured at different incident angles from 0° to 60°, as shown in Figure 6b. In general, as the angle of incidence (AOI) increased, the absorption band narrowed. When the AOI was increased from 0° to 45°, the average absorbance of the surface with Ag NPs was 98.7%, 99.1%, and 99.3%, respectively. The nanostructures exhibited near-perfect superabsorption with increasing AOI. However, in the case of 60° AOI, the absorption of the nanostructures decreased at wavelengths above 1600 nm; this may have been because the energy of the structure with Ag NPs was mainly concentrated on the upper position of the structures when the wavelength was greater than 1600 nm, and the energy was easily radiated into the air in this mode. Thus, the proposed structure is suitable for broadband and wide-angle absorption applications, which providing large angle detection and measurement for optical systems in atmospheric and remote sensing.

### 3.3. The Effect of Randomly Distributed Nanostructured Si Surface on Absorption

Firstly, the effect of random diameters of Ag NPs on the optical properties of Si metasurfaces was investigated, and the reference diameter $Ds$, height $H$ and period $P$ were set to 0.5 µm, 0.9 µm and 0.8 µm, respectively. In order to simulate the random structures in FDTD, the periodic arrangement of the structure array was replaced by the random diameter arrangement, the variation in the diameters of Ag NPs within each unit cell was set as the deviation value $\Delta d$, and $\Delta d$ is set as 0 µm, 0.1 µm, 0.2 µm, and 0.3 µm, respectively. For example, when $\Delta d$ = 0.1 µm, the diameter selection range of the Ag NPs was 0.5 µm ± 0.1 µm. Figure 7 shows the absorbance curves of different diameter deviation values $\Delta d$ as a function of wavelength. It can be seen that when $\Delta d$ was less than 0.2 µm, the absorbance of the structures was less affected, which was similar to the periodic structures ($\Delta d$ = 0 µm). However, when $\Delta d$ = 0.3 µm, the absorbance of the structures decreased significantly. This was because, with the increase in diameter fluctuation, the volume of the Ag NPs changed significantly, which changed the resonance mode between Ag NPs, and the light coupled into the structure decreased.

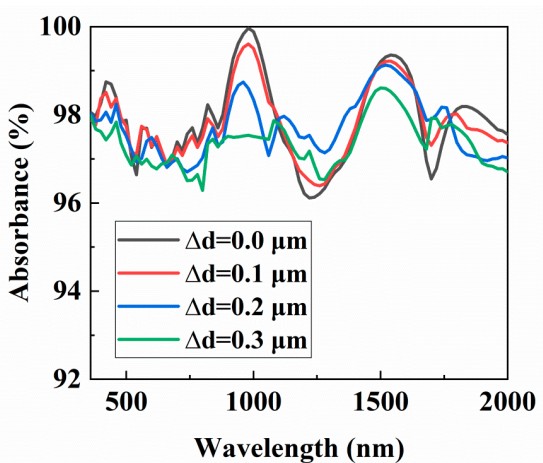

**Figure 7.** The measured absorbance curves of different diameter deviation values $\Delta d$ as a function of wavelength.

Similarly, Figure 8 depicts the absorbance curves of different structural height deviation values $\Delta h$ as a function of wavelength. $\Delta h$ was chosen as 0 µm, 0.1 µm, 0.2 µm, 0.3 µm, and 0.4 µm, respectively. We can observe that $\Delta h$ was negatively correlated with the absorption rate; that is, the larger the variation of $\Delta h$, the smaller the absorbance of the random structures. When $\Delta h$ = 0.4 µm, the average absorbance of the random structures was reduced to 97.3%. Moreover, compared with the periodic structures ($\Delta h$ = 0 µm), with the increase in the height deviation value $\Delta h$, the resonance peak also changed, resulting in

a red-shift phenomenon. This may have been because with the increase in Δh, if the structure as subdivided into N layers, the refractive index of each layer was no longer a smooth gradient change, resulting in enhanced reflection of incident light and reduced absorption.

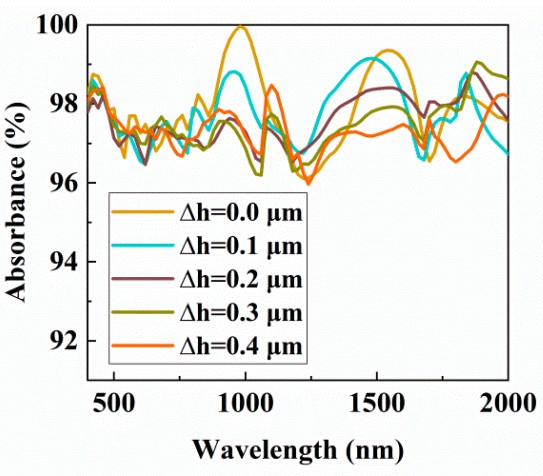

**Figure 8.** The measured absorbance curves of different height deviation values Δh as a function of wavelength.

In addition, the effect of different positional deviation values Δp on Si metasurface absorption is shown in Figure 9. It can be seen that the change of Δp barely changed the incident light absorbance of the random structures. Only when Δp = 0.15 μm, the absorption of the structure decreased slightly around the wavelength of 1000 nm. For changes in Δp, the average period of the structures was constant for the entire simulated array, since each structure was displaced within its own unit cell. Therefore, when Δp fluctuated within a certain range, the absorption effect of random arrays and periodic arrays on the Si metasurface was consistent.

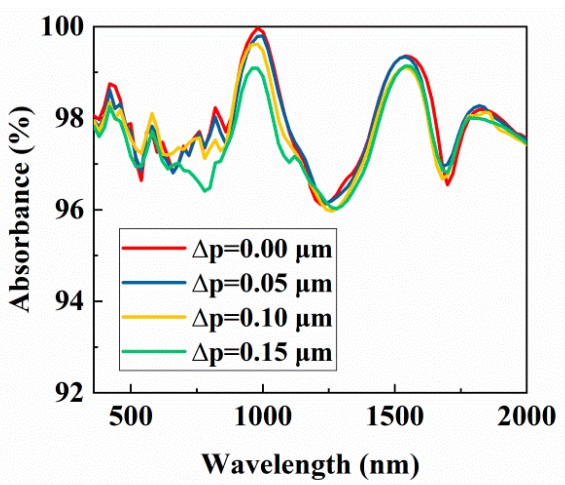

**Figure 9.** The measured absorbance curves of different positional deviation values Δp as a function of wavelength.

### 3.4. Superabsorption Analysis of Nanostructured Si Metasurfaces

Furthermore, we investigated the effect of Ag NPs on the absorption properties of Si metasurface and discussed the electric field distribution. Three wavelengths of incident light (0.62 μm, 1.05 μm, and 1.73 μm) were selected for the electric field distribution patterns of Ag NPs on Si substrate (before ICP process), the nanostructures without and with Ag NPs, as shown in Figure 10. Figure 10a–c describe the electric field intensity distribution of

the cross-section of Ag NPs on Si substrate in the x-z direction. Meanwhile, Figure 10d–f and g–i show the nanostructures without and with Ag NPs.

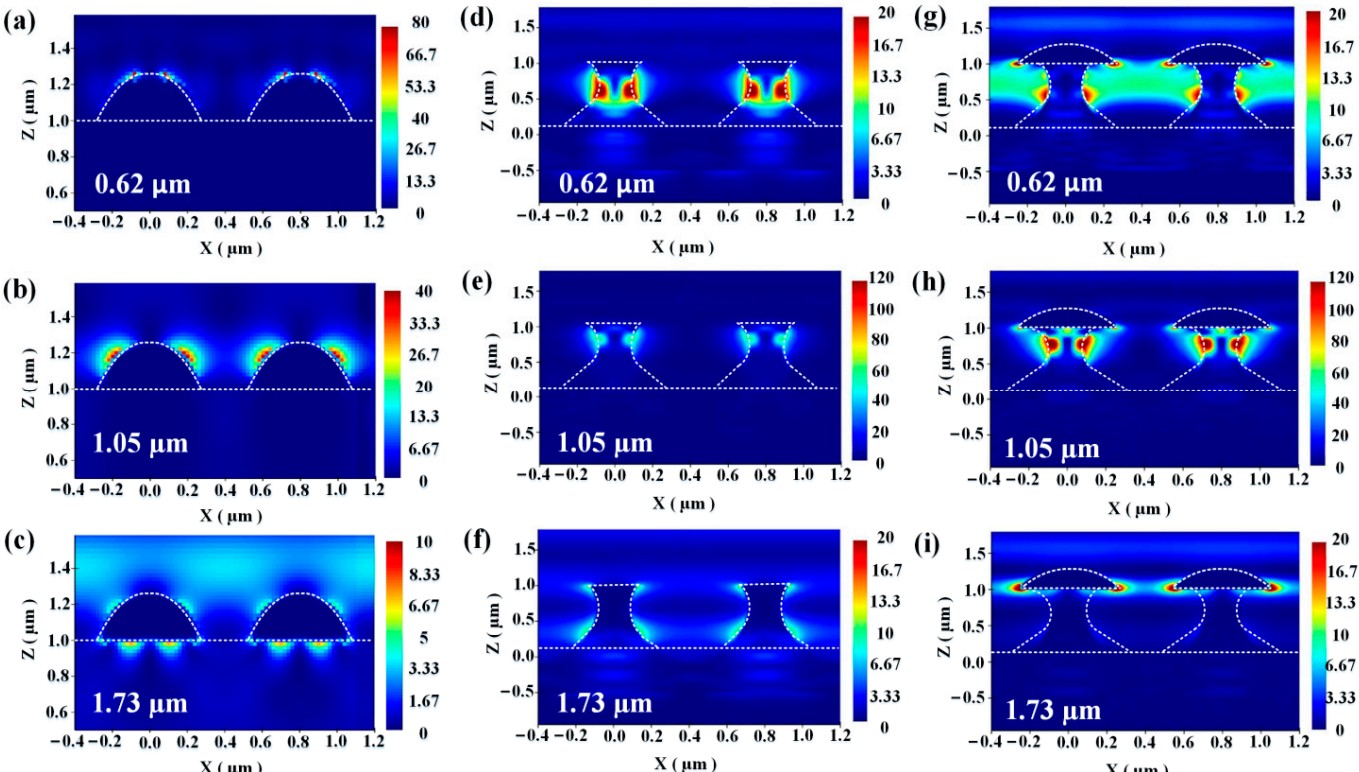

**Figure 10.** The measured electric field distribution on adjacent Ag NPs on Si substrate (**a–c**) and nanostructures without Ag NPs (**d–f**) and with Ag NPs (**g–i**) at different incident wavelengths: 0.62 μm, 1.05 μm, and 1.73 μm (Scales are $|E|^2$ ).

The images in Figure 10a–c show the three electric field modes corresponding to individual Ag NPs at different wavelengths, which contribute to the enhanced absorption. At 0.63 μm, localized surface plasmon resonance (LSPR) mainly occurred on the surface of NPs, as shown in Figure 10a. Figure 10b shows that the electric field was mainly distributed between the NPs and air at 1.05 μm, and this mode could emit energy into space with increased scattering. Moreover, a part of the LSPR was coupled between the NPs and the Si substrate at 1.73 μm, as shown in Figure 10c, while for the nanostructures without Ag NPs, the energy was mostly concentrated near the waist of the structure, and its absorption mainly depended on the structured silicon. The FP resonance was formed on the side surface of the Si absorber to absorb the incident light. Since the sidewall profile of the Si absorber was a quasi-smooth curved surface, the incident light was strongly absorbed, as shown in Figure 10d–f. On this basis, combining Ag NPs with Si absorber, it can be seen from Figure 10g,h that the LSPR effect of Ag NPs enlarged the range of the FP cavity about 10 times, and the electric field strength $|E|^2$ increased about four times. In addition, at the wavelength of 1.73 μm, the high-order LSPR of Ag NPs was excited, increasing the total absorption of the structure.

## 4. Conclusions

Based on the simulation model, we used a FDTD software to optimize the structural parameters of a $3 \times 3$ unit cell. Then, a silicon metasurface with mushroom-nanostructures was fabricated using thermal annealing and ICP processes. The average absorbance of the nanostructured Si metasurface with Ag NPs was 98.7% in the wavelength range of 400–2000 nm and reached up to approximately 99% for incident angles of 0–45°. Meanwhile, it was proved by FDTD simulation that random structure and periodic structure have the

same effect on Si metasurface absorption to a certain extent. From the analysis of the electric field distribution of the desired structures at different wavelengths, it was proved that the addition of Ag NPs excites the resonance between the particles, so that the broadband absorption of the mushroom-nanostructured Si metasurface is improved. This is expected to benefit the development of devices such as photovoltaic cells and photoelectric detectors.

**Author Contributions:** Conceptualization, J.W. and Y.F.; methodology, J.W.; software, J.W.; validation, J.W. and B.Y.; formal analysis, J.W.; investigation, J.W. and B.Y.; resources, M.O. and B.Y.; data Curation, M.O. and B.Y.; writing—original draft preparation, J.W.; writing—review and editing, M.O. and Y.F.; supervision, Y.F.; project administration, Y.F.; funding acquisition, M.O. and Y.F. All authors have read and agreed to the published version of the manuscript.

**Funding:** This research was funded by the National Natural Science Foundation of China (61705018); the Foundation for local scientific and technological development guided by the central government of China (202002037jc); Science and technology research project of Education Department of Jilin Province (JJKH20210814KJ); and the 111 Project of China (D17017).

**Institutional Review Board Statement:** Not applicable.

**Informed Consent Statement:** Not applicable.

**Data Availability Statement:** Not applicable.

**Acknowledgments:** The authors thank the Key Laboratory of Advanced Optical System Design and Manufacturing Technology of the universities of Jilin Province for their SEM technical support.

**Conflicts of Interest:** The authors declare no conflict of interest.

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
