# Peer review of "Simulation of Mushroom Nanostructures with Ag Nanoparticles for Broadband and Wide-Angle Superabsorption"

_coatings, doi:10.3390/coatings12081208_

Round 1
Reviewer 1 Report
[General Comments]
The authors fabricated silicon mushroom nanostructures with Ag caps. They evaluated the absorption properties by both experiments and simulations. The nanostructures showed very large absorption efficiency across broad wavelength regions. Also, the simulation well reproduced the experimental results. They also discussed the absorption enhancement mechanism. However, I am afraid I have to disagree with their conclusion. Also, their interpretation is still too phenomenological.
Therefore, I think a significant revision is needed to bring this study to the level required for publication in Coatings.
[Specific Comments]
1. The authors mention that the physical mechanism behind the superabsorption has not been explained previously (lines 52-53). Also, they mention the mechanism in section 3.4.
However, I am afraid I have to disagree with their conclusions. For example, they claim "it was proved that the addition of Ag NPs excites the resonance between the particles, so that the broadband absorption of the mushroom-nanostructured Si metasurface is improved." (lines 253-354). In this case, what does absorb the electric near field? A lower reflectance (or higher absorption) results from the materials' electronic/vibrational excitation. Additionally, there is no relationship between the spatial distribution of the near field and absorption frequency. Furthermore, the authors do not explain the angle dependence discussed in Figures 6 and 7.
First, the authors should clearly evaluate the effect of the LSPR alone on absorption efficiency. After that, they should discuss the regime of the interaction between the LSPR and Si absorber (Just liner combination, weak or strong coupling, etc.)
2. The optical properties of Au-NDs on flat Si (the sample before ICP processes) should be evaluated to understand LSPR modes without the effect of Si nanostructure.
3. Figures 6 and 7: The graph area at a longer wavelength region than 2000 nm should be trimmed. Additionally, plot color should be unified between Figs 6 and 7.
4. Figures 8-11: It should be clearly described in captions that these are simulated results.
5. Figure 11: There is a lack of the parameter of the intensity. Is this |E/E0| or |E/E0|^2?
6. Section 2.2: The authors should describe the simulation detail. For example, how about the mesh size? Which index parameters are employed?
Additionally, the structure should be uniquely defined. For example, the waist shape and position of the structured Si should be represented using a geometric method.
Reviewer 2 Report
In the article entitled “Analysis and simulation of mushroom nanostructures with Ag nanoparticles for broadband and wide-angle superabsorption”, Wu and co-workers study the effect of Ag nanoparticles on the absorption in mushroom-like nanostructured Si metasurfaces. Due to the excitation of the Ag NPs resonance between particles, the average broadband absorption of the metasurface is improved from 90.8% to 98.7%. It could be technologically exploited in photovoltaic cells and photoelectric detectors. The manuscript is well organized and shows new simulation studies about the mushroom-like metastructures that supports the conclusions. A minor revision of the paper must be carried out before reconsidering its publication. The following points should be addressed:
1) Authors observe an increment of the absorbance in the metasurfaces with silver from 90.8% to 98.7%. In the abstract, it is indicated that an 8% is increased in the absorbance, but it is not the same to increase the 8% from 30% to 38% than the increment from 90.8% to 98.7%. I suggest that you indicate the average values of the absorption in the introduction.
2) In previous recent work, authors fabricated the same metasurfaces and studied them also using the finite difference time domain (FDTD) method, comparing experimental and simulated results. In the present work, the authors indicated that in the previous work “the physical mechanism behind this superabsorption has not been reasonably explained”, and in the present article, the absorption of the metasurfaces has been analyzed using (FDTD). However, in the previous article, important conclusions about the increment of the absorption of these structures with the Ag nanoparticles were obtained. I suggest that the authors summarize the results of the previous work in the Introduction section to clarify the novelty and the relevance of the present paper.
3) The study of the electric field distribution on adjacent nanostructures is very revealing because it not only indicates that the Ag NPs increase the absorbed energy of the incident light, but also how the distribution of the surface plasmon resonance modes changes, and the dependence of these changes on the incident wavelengths. In my opinion, it should be highlighted in the abstract, and these results should be discussed with more detail in the manuscript.
Reviewer 3 Report
In the given manuscript, Jinshuang Wu et. al. studied mushroom-nanostructured Si metasurface for superabsorption which the authors previously have reported. The Si metasurface consisting of Ag nanoparticles on the waist-shaped Si nanostructures and Si substrate was fabricated by metal thermal annealing and inductively coupled plasma process. The metasurface was analyzed to reveal its high absorption in broadband wavelength and under wide incident angle by finite difference time domain method (Lumetical). The authors firstly presented simulation results with structural parameters taken from experimental structures and showed the effect of the parameter change. The simulation results showed good agreement with the experimental one and especially highlighted strong energy concentration in the electric field distribution related to the superabsorption of the metasurface. I'd like to recommend the submitted manuscript to publish in Coatings after minor improvements. Below are my questions and comments.
1. In my opinion, every published paper has to contain an analysis of something, therefore the word 'analysis' is unnecessary in a paper's title. It can make the title more compact.
2. In the suggested structure, doesn't the curvature of waist matter? In other words, the position and diameter of the thinner part of Si nanostructure seem to be important parameter but was not mentioned.
3. Fig 1(c) seems to be unnecessary.
4. In Lumerical simulation, where is the transmittance monitor to calculate transmittance (T) placed? Is the transmittance for 1) Ag nanoparticles and waist-shaped Si nanostructures or 2) Ag nanoparticles, Si nanostructures, and Si substrate? If the former, it is not possible to compare the simulated T with the measured T, as well as absorbance A. If the latter, it would be better to mention the thickness of the substrate.
5. In line 82 on page 2, the authors used the periodic boundary conditions in the x and y axis. But is it correct to use periodic boundary conditions for oblique incident light, such as 15 and 60 degrees?
6. It would be better to mention the full word of RCA in line 90 on page 3.
7. The simulated absorbance in Fig. 6 shows good agreement with the measured one in Fig. 7. for incident angle of 0 to 45 degree while for 60 degree the simulated one cannot support the measured one well for the longer wavelength. The authors need to note the reason for the large difference above 1600 nm.
8. In my opinion, Fig. 7 would be better to be Fig. 6(b) for easy comparison.
9. In section 3.3: in line 186 on page 6, why do the authors choose the reference parameter Ds of 0.5 um and H of 0.9 um instead of 0.6 um and 1.0 um used in the previous section? The authors need to mention the reason. And is it the reason that the line for d=0.0 um in Fig. 8 is different from the absorbance line in Fig. 3(b)?
10. In line 237 on page 8, the authors mentioned that the Ag particles mainly absorb as shown in Fig. 11(d)-(f). But Fig. 11(e) shows the strong energy concentration rather in Si waist, therefore at 1.05 um wavelength the main origin of superabsorption seems not Ag nanoparticles. The authors could numerically compare the absorbance portion of Ag nanoparticles, Si nanostructure, and Si substrate.
11. In line 240 on page 8, the authors mentioned 'comparing Figures 11(b)-(d) with Figures 11(e)-(f)' to see the effect of Ag nanoparticles. I think there seems to be a typo. Is it correct?
Round 2
Reviewer 1 Report
The authors addressed to reviewers’ comments point by point and revised the manuscript.
I would suggest that the manuscript will be published in Coating.